# Analysis of the Biomarkers for Neurodegenerative Diseases in Aged Progranulin Deficient Mice

**DOI:** 10.3390/ijms23020629

**Published:** 2022-01-06

**Authors:** Xiangli Zhao, Sadaf Hasan, Benjamin Liou, Yi Lin, Ying Sun, Chuanju Liu

**Affiliations:** 1Department of Orthopaedic Surgery, New York University Medical Center, New York, NY 10003, USA; Xiangli.Zhao@nyulangone.org (X.Z.); Sadaf.Hasan@nyulangone.org (S.H.); 2The Division of Human Genetics, Cincinnati Children’s Hospital Medical Center, Cincinnati, OH 45229, USA; Benjamin.Liou@cchmc.org (B.L.); Yi.Lin@cchmc.org (Y.L.); ying.sun@cchmc.org (Y.S.); 3Department of Pediatrics, University of Cincinnati College of Medicine, Cincinnati, OH 45229, USA; 4Department of Cell Biology, New York University School of Medicine, New York, NY 10016, USA

**Keywords:** Progranulin, neurodegenerative disease, brain, inflammation, biomarker

## Abstract

Neurodegenerative diseases are debilitating impairments that affect millions of people worldwide and are characterized by progressive degeneration of structure and function of the central or peripheral nervous system. Effective biomarkers for neurodegenerative diseases can be used to improve the diagnostic workup in the clinic as well as facilitate the development of effective disease-modifying therapies. Progranulin (PGRN) has been reported to be involved in various neurodegenerative disorders. Hence, in the current study we systematically compared the inflammation and accumulation of typical neurodegenerative disease markers in the brain tissue between PGRN knockout (PGRN KO) and wildtype (WT) mice. We found that PGRN deficiency led to significant neuron loss as well as activation of microglia and astrocytes in aged mice. Several characteristic neurodegenerative markers, including α-synuclein, TAR DNA-binding protein 43 (TDP-43), Tau, and β-amyloid, were all accumulated in the brain of PGRN-deficient mice as compared to WT mice. Moreover, higher aggregation of lipofuscin was observed in the brain tissue of PGRN-deficient mice compared with WT mice. In addition, the autophagy was also defective in the brain of PGRN-deficient mice, indicated by the abnormal expression level of autophagy marker LC3-II. Collectively, comprehensive assays support the idea that PGRN plays an important role during the development of neurodegenerative disease, indicating that PGRN might be a useful biomarker for neurodegenerative diseases in clinical settings.

## 1. Introduction

Neurodegenerative diseases have become a growing threat to human health. Examples of neurodegenerative diseases are Parkinson’s disease (PD), Alzheimer’s disease (AD), Huntington’s disease and frontotemporal dementia (FTD). [1]. Protein abnormalities are one of the main characteristics of neurodegenerative diseases [2]. One way to learn about how these diseases work is to develop an effective model and identify helpful biomarkers of disease. PGRN, a growth factor-like molecule, is involved in the regulation of wound healing, cancer progression, lysosomal storage disease and neurodegenerative diseases [3,4,5,6,7,8]. The landmark study about the involvement of PGRN in neurodegenerative disease showed its association with FTD in 2006 [9]. The heterozygous mutation of *GRN* leads to haploinsufficiency of PGRN, which is the major cause of FTD, characterized by the aggregates of ubiquited-protein in TDP-43 [9]. However, the homozygous *GRN* mutations result in neuronal ceroid lipofuscinosis (NCL), a lysosomal storage disease [10]. The abnormalities in the CNS of *Grn*^−/−^ mouse models facilitate the understanding of PGRN’s role in neurodegenerative diseases. Studies have revealed that the brains of *Grn*^−/−^ mice display gliosis, lipofuscin deposits and premature death, and *Grn*^+/−^ mice display age-dependent defects in emotional behavior and social recognition [11,12,13]. In response to injury, PGRN-deficient mice showed more neuron loss and increased microgliosis in the CNS [14]. PGRN has also been implicated in AD, both in human AD cases and in animal models, where increased expression of PGRN occurs in activated microglia [15]. In addition to the link of *Grn*’s mutation to the development of FTD and NCL, PGRN represents a protective role in the growth and development of neurons in various animal models [16]. Furthermore, gene delivery of PGRN not only exhibits protective effects on dopaminergic neurons but also improves lysosomal dysfunction and microglial pathology in a mouse model of PD, FTD and NCL [17,18,19]. In addition, PGRN is a well-established regulator of autophagy [20,21,22].

In the current study, the effect of PGRN deficiency on CNS was analyzed systematically. We found that PGRN deficiency led to the activation of microglia and astrocytes as well as the loss of neurons in aged mice. Moreover, typical neurodegenerative markers, namely α-syn, TDP-43, Tau and β-amyloid, were all accumulated in the brain of PGRN-deficient mice as compared to WT mice. In addition, PGRN-deficient mice showed enhanced aggregation of lipofuscin in the brain compared with WT mice. Furthermore, PGRN deficiency resulted in defective autophagy in the brain.

## 2. Results

### 2.1. PGRN Deficiency Led to Neuron Loss

Neurodegenerative disorders are characterized by progressive neuronal loss and deposition of dysfunctional proteins [2]. To examine the physiological function of PGRN in the CNS, we leveraged PGRN knockout mice (*Grn*^−/−^) (Appendix A) maintained in the laboratory [23,24,25,26]. The brain tissues from 12 m WT or *Grn*^−/−^ mice were collected and then used to analyze the maturation of the neurons. As a growth-factor-like molecule, PGRN acts as a neurotrophic factor stimulating neurite outgrowth [27], hence, loss of PGRN impairs appropriate neurite outgrowth and branching [28,29]. To investigate the effect of PGRN deficiency on neuron development, we tested the expression level of a neuron marker, NeuN, at different ages. As shown in Figure 1A–C, we found that the deficiency of PGRN did not affect the NeuN expression level at the age of one month. However, the NeuN expression level was decreased at four months and eight months with a dramatic loss in the 12 month brain, mainly affecting the midbrain (Figure 1D,E).

### 2.2. Ablation of PGRN Promoted the Activation of Microglia and Astrocyte in Aged Mice

Microglia and astrocytes are two main types of neuroglia, which are the predominant non-neuronal cells in the CNS [30]. To determine the effect of PGRN on the activity of microglia and astrocytes, Iba1 and GFAP, the markers of activated microglia and astrocytes, respectively, were used to perform analysis [31,32]. Iba1 positive cells were found to be critically accumulated in different regions of the brain of *Grn*^−/−^ mice including cortex, midbrain, and hippocampus in comparison with WT mice (Figure 2 and Appendix A). Similarly, GFAP accumulation was intensified in the midbrain and cerebellum of *Grn*^−/−^ mice compared to WT mice (Figure 3). Hence, these observations suggested that PGRN deficiency led to severe microglia and astrocyte activation in the CNS.

### 2.3. PGRN Deficiency Induced Accumulation of Phosphorylated α-Syn in Aged Mice

Protein abnormalities are one of the main characteristics of neurodegenerative diseases. α-synuclein (α-syn) is developmentally expressed in the brain and plays a driving role in neurodegenerative comorbidities [33,34]. A growing number of studies report that the phosphorylated α-syn at residue S129 was aberrantly accumulated in the brain of PD patients [35,36]. Herein, we analyzed the distribution of phospho-S129-α-syn in the aged-brain of WT and *Grn*^−/−^ mice. As shown in Figure 4, dramatic accumulation of α-syn (p-S129) was found in the brain of *Grn*^−/−^, specifically in the hippocampus, cerebral cortex, and the midbrain, which was not observed in the brain of WT mice.

### 2.4. Increased Accumulation of TDP-43 in Aged PGRN-Deficient Mice

TDP-43 is another protein that has been robustly linked to the pathogenesis of several neurodegenerative disorders [37]. More than 90% of patients with sporadic amyotrophic lateral sclerosis and about 50% of frontotemporal lobar degeneration (FTLD) patients exhibit widespread abnormality of TDP-43 [38,39]. In the aged PGRN-deficient mice, we found that the fluorescence intensity of TDP-43 was strongly elevated in the brain tissue when compared to the age-matched WT mice (Figure 5). These observations confirmed that PGRN deficiency potentiates the aggregation of TDP-43 in the CNS.

### 2.5. PGRN Deficiency Induced Tau Aggregation in the CNS of Aged-Mice

Tau, a microtubule-associated protein, is predominantly expressed in the neurons and involved in many neurodegenerative diseases such as AD, frontotemporal dementia with parkinsonism-17 (FTDP-17), and PD [40,41,42]. Hyperphosphorylated aggregated tau is one of the pathological hallmarks of AD [43]. Furthermore, reduced tau expression in FTLD or FTD was reported to be associated with PGRN mutation [40,44]. Therefore, to determine the effect of PGRN deficiency on tau aggregation, the distribution and protein level of tau was examined in the brain tissue of WT and *Grn*^−/−^ mice using immunofluorescence. We found that tau protein was elevated mainly in the midbrain of *Grn*^−/−^ mice as compared to age-matched WT mice, with no significant difference in other regions of the brain (Figure 6). Hence, these results suggest that elevated tau only accounts for a small part of the whole brain of *Grn*^−/−^ mice.

### 2.6. β-Amyloid Accumulated in Aged PGRN-Deficient Mice

AD is the most common type of dementia, and its pathogenesis is widely believed to be driven by the production and deposition of β-amyloid peptides [45,46]. Loss function mutations in PGRN are considered to be a risk factor for AD [47]. Previous research on the effect of PGRN on β-amyloid deposition is inconsistent [48,49,50]. In the current study, we found that β-amyloid is mainly deposited in the midbrain of *Grn*^−/−^ mice, which was not observed in the age-matched WT mice (Figure 7). The expression of β-amyloid in other brain regions, such as hippocampus and cortex, was comparable between WT and *Grn*^−/−^ mice (Figure 7). These observations suggest that PGRN deficiency selectively affects the deposition of β-amyloid in different regions of the brain.

### 2.7. PGRN Deficiency Results in the Accumulation of Lipofuscin and Defective Autophgy

Age is the best-known risk factor for various neurodegenerative diseases and one of the most striking morphologic changes in neurons during aging is the presence of lipofuscin aggregates [51]. To determine the effects of PGRN deficiency on lipofuscin accumulation, we collected the 17-month-old brain tissue from WT and *Grn*^−/−^ mice and compared lipofuscin accumulation in their brain tissues. As shown in Figure 8, unlike in WT mice, we found a substantial accumulation of lipofuscin in the brain tissue of *Grn*^−/−^ mice, which suggested a critical role of PGRN in aging-associated diseases, particularly neurodegenerative disorders.

In addition, our previous results showed that PGRN deficiency led to the defect of autophagy in Gaucher disease, a kind of lysosomal disease [20]. Therefore, we also measured the well-known autophagy related markers LC3-II, and we found that LC3-II level was increased in PGRN-deficient mice (Appendix A), indicating involvement of PGRN in autophagy processes in the brain.

## 3. Discussion

PGRN haploinsufficiency is associated with neurodegenerative diseases and is widely studied in FTLD and NCL. However, most studies were associated with pathological conditions, such as in *Grn* mutant-FTL patients, *Grn* mutant-AD, or PD. In this study, we comprehensively compared the expression pattern of different brain cell types and the well-known neurodegenerative markers in the brain tissues between WT and *Grn*^−/−^ mice, demonstrating the crucial effects of PGRN deficiency on neurodegenerative development.

Given that PGRN functions as a growth-factor-like molecule, we first checked the effect of PGRN deficiency on neuron outgrowth. The results showed that the PGRN deficiency affected the outgrowth of neurons starting from 4 months; however, no effect was observed at 1 month of age. PGRN deficiency was reported to decrease the neuronal arborization, length and neurite outgrowth and PGRN treatment was shown to enhance the neurite outgrowth [27]. Consistently, our results supported the role of PGRN as a neurotrophic factor.

Microglia, the resident macrophages, are the only endogenous immune cells within the CNS [52]. The systemic inflammation associated with a chronic neurodegenerative disease is mainly mediated by the activation of these cells [53,54]. In addition to microglia, astrocytes are the largest and most abundant glial cells in the CNS, contributing to the maintenance of its health and function, which have also been implicated in the onset and progression of several neurodegenerative diseases [55,56]. Recently, we reported that PGRN mediated the macrophage polarization and switch [57], suggesting the critical role of PGRN in macrophage-mediated inflammation. In this study, we revealed that the brains of PGRN-deficient mice displayed enhanced activation of microglia and astrocytes compared with age-matched WT mice, further indicating the critical role of PGRN in regulating systematic inflammation in CNS.

Neuroinflammation is a key paradigm in the pathogenesis of chronic neurodegenerative diseases [58]. Therefore, we next investigated the disease progression by measuring several widely known characteristic neurodegenerative markers [34,39,59,60]. Alpha-synuclein (α-syn) plays a fundamental role in the pathogenesis of neurodegenerative synucleinopathies, and pathologic α-syn was reported to be associated with PGRN mutation-related disease [34,35]. TDP-43 positive inclusions are prime pathologic features in both amyotrophic lateral sclerosis (ALS) and FTD [61]. A substantial proportion of patients with ALS and FTLD exhibit TDP-43-positive neuronal inclusions [39]. The neurotoxic microglia promote TDP-43 proteinopathy in PGRN-deficient mice [62]. In vitro, the suppression of PGRN expression leads to the formation of TDP-43 inclusions [63]. Our current data corroborated with previous studies, shown by the evident accumulation of TDP-43 in the whole brain tissue of PGRN-deficient mice which was not observed in WT mice.

Tauopathies-associated neurodegenerative diseases are hallmarked by the aggregation of tau or phosphorylated-tau including AD, FTDP-17 and corticobasal degeneration (CBD) [64]. Herein we found that PGRN deficiency also led to tau aggregation in specific regions of the brain, further indicating the association of PGRN with neurodegenerative disorders. In addition, neurodegenerative disorders are also characterized by the deposition of misfolded proteins, which are frequently found in a β-sheet-rich fibrillar protein conformations known as amyloids [60]. In the current study, PGRN deficiency was shown to increase the deposition of β-amyloid, consistent with previous reports where PGRN administration could attenuate β-amyloid deposition [48,50]. Other studies have also shown that insufficient PGRN reduced β-amyloid deposition in amyloid precursor protein (APP) transgenic mice [49]. This discrepancy might be explained by using of different mice models and/or strains.

Moreover, PGRN deficiency was also found to promote lipofuscin aggregation in the brain of 17-month-old mice compared with the age-matched WT mice, indicating the involvement of PGRN in aging. In addition, our previous study found that PGRN is involved in autophagy through mediating the autophagosome-lysosome fusion [20], and so we also checked the expression pattern of autophagy marker LC3 in the brain tissue from WT and *Grn*^−/−^ mice. As anticipated, the LC3-II level was increased significantly in *Grn*^−/−^ mice compared with WT mice (Appendix A), suggesting the deficiency of PGRN impaired the autophagy process.

In conclusion, our current data demonstrated that PGRN deficiency led to neuron loss as well as activation of microglia and astrocytes, which induced inflammatory response in the brain. Additionally, PGRN deficiency accumulated typical neurodegenerative markers, including α-syn, TDP-43, Tau, and β-amyloid, suggesting the involvement of PGRN in neurodegenerative disorders. Furthermore, the higher aggregation of lipofuscin in PGRN-deficient mice indicated the involvement of PGRN in aging. Hence, these data further validate a critical role of PGRN during the development of neurodegenerative diseases, supporting its role as a potential diagnostic biomarker for neurodegenerative diseases.

## 4. Materials and Methods

### 4.1. Mice

All animal experiments were approved by the Institutional Animal Care and Use Committee (IACUC) of New York University School of Medicine and Cincinnati Childern’s Hospital Medical Center. Mice were group housed within the rodent barrier facility at Skirball Institute of Biomolecular Medicine with ad libitum access to food and water in a specific pathogen-free room under controlled temperature and humidity on a 12 h light/dark cycle. C57BL6/J background WT and *Grn*^−/−^ mice were acquired from Jackson Laboratory and lines were maintained within the animal housing facilities.

### 4.2. Reagents and Materials

The following antibodies were used in this study: antibodies against β-amyloid (MOAB-2) were purchased from Novus Biologicals (Littleton, CO, USA). Antibodies against α-synuclein (phospho S129) (ab51253), Iba1 (ab178846), and NeuN (ab177487) were purchased from Abcam (CA, UK). Antibodies against Tau-5 (MA5-12808), GFAP (PA5-16291), and PGRN (40-3400) were purchased from Thermo Fisher Scientific (Bridgewater, NJ, USA). Antibody against TDP-43 (10782-2-AP) was purchased from Proteintech (IL, USA). Antibodies against LC3B (2775S) and GAPDH (2118) were purchased from Cell Signaling Technology (Danvers, MA, USA). Fluorescence-labeled secondary antibodies were purchased from Jackson ImmunoResearch Laboratories, Inc. (West Grove, PA, USA). DAPI (H-1200) was purchased from VECTOR Laboratories (Burlingame, CA, USA). RNeasy Mini Kit (74104) was purchased from Qiagen (MD, USA). High-Capacity cDNA Reverse Transcription Kit (43-688-14) was purchased from Applied Biosystems™ (Waltham, MA, USA). SYBR™ Green PCR Master Mix (4309155) was purchased from Thermo Fisher Scientific (Waltham, MA, USA).

### 4.3. Tissue Preparation for Western Blot Analysis

Mice were sacrificed using sodium pentobarbital. Required tissues were dissected after transcardial perfusion with saline and snap frozen in dry ice. Before the experiment, the frozen tissues were homogenized with bead homogenizer and lysed with cold RIPA buffer. After sonication and centrifugation, supernatants were collected and prepared for Western blotting analysis.

### 4.4. Immunofluorescence Staining (IF)

Perfused brains were fixed in 4% paraformaldehyde and processed for tissue blocks. Frozen mice brain sections were fixed with 4% paraformaldehyde for 10 min, followed by washing with PBS. The fixed tissues were permeabilized by 0.3% TritonX-100 for 15 min and then washed with PBS. After permeabilization, the tissues were blocked with 1:50 dilution of normal donkey serum for 1 h, followed by incubation with primary antibodies at 4 °C overnight. The next day, slides were washed with PBS and then incubated with indicated fluorescence-labeled secondary antibodies for 1 h at RT. After washing with PBS, the tissues were mounted on an anti-fade medium containing DAPI. The images were taken by Leica TCS SP5 confocal system and quantification was analyzed using Image J.

### 4.5. Immunohistochemistry Staining (IHC)

Mice brain tissues were fixed in 10% formalin and embedded in paraffin. For immunohistochemistry staining, deparaffinized and hydrated sections were incubated with 0.1% trypsin for 30 min at 37 °C. Then the sections were incubated with indicated antibodies at 4 °C overnight. Detection was performed using the Vectastain Elite ABC kit (Vector) and the positive signal was visualized with 0.5 mg/mL 3,3-diaminobenzidine in 50 mM Tris-Cl substrate (Sigma-Aldrich, St. Louis, MO, USA) and counterstained with 1% methyl green. Images were acquired with a Zeiss microscope. The positive cell number was counted manually. The quantification was done by the mean value of positive cell number ratio between WT mice and *Grn*^−/−^ mice.

### 4.6. Transmission Electron Microscope (TEM)

WT and *Grn*^−/−^ mice were anesthetized, and the brain samples were fixed in 1% OsO4 for 1 h, followed by block-staining with 1% uranyl acetate for another hour. Next, these samples were dehydrated and finally embedded in Embed 812 (Electron Microscopy Sciences, Hatfield, PA, USA). Then, 60 nm sections were cut and stained with uranyl acetate and lead citrate by standard methods. Stained grids were examined under Philips CM-12 electron microscope (FEI; Eindhoven, The Netherlands) and photographed with a Gatan (4 k × 2.7 k) digital camera (Gatan, Inc., Pleasanton, CA, USA). The preparation of these cell samples for TEM was done with the assistance of Dr. Fengxia Liang at NYU Medical School OCS Microscopy Core.

### 4.7. Individual Quantitative RT–PCR

Total RNA was extracted from lung, brain, and liver of WT mice or *Grn*^−/−^ mice with the RNeasy Mini Kit. cDNA was prepared using 1 μg RNA with the High-Capacity cDNA Reverse Transcription Kit. SYBR green-based quantitative PCR was performed in triplicate using mouse primers to Grn and Gapdh (Real-Time PCR System, Applied Biosystems). mRNA levels were normalized to Gapdh and reported as relative mRNA fold change.

### 4.8. Statistical Analysis

Unpaired Student’s *t*-test was performed for the comparison between WT and Grn^−/−^ mice. All statistical analysis was performed using GraphPad Prism 7 Software. Data are shown as mean ± SD, * *p* < 0.05, ** *p* < 0.01.

## Figures and Tables

**Figure 1 ijms-23-00629-f001:**
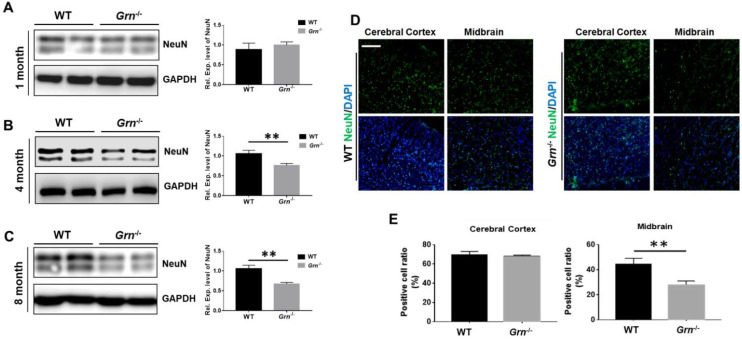
PGRN deficiency led to neuron loss. Brain tissues were collected from 1-month-, 4-month-, 8-month-, and 12-month-old WT mice or *Grn*^−/–^ mice. 1-, 4-, and 8-month-old brain tissues were homogenized and lysed in lysis buffer, and then the tissue lysate was used to perform Western blotting. The 12-month-old brain tissues were sectioned and used for immunofluorescence staining. (**A**), (**B**), (**C**) Western blot analysis of NeuN in the 1-, 4-, and 8-month-old brain tissue of WT and *Grn*^−/–^ mice. (**D**) Immunofluorescent staining of NeuN in 12-month-old brain tissue of WT or *Grn*^−/–^ mice. (**E**) Statistical analysis of (**D**). Scale bar: 100 µm. *n* = 3. Data presented as mean ± SD. ** *p* < 0.01.

**Figure 2 ijms-23-00629-f002:**
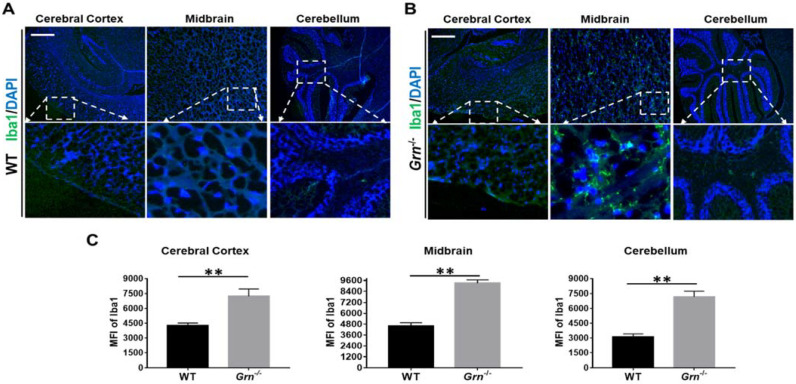
Ablation of PGRN promoted the activation of microglia in aged mice. Brain sections from 12-month-old WT and *Grn*^−/–^ mice were stained with anti-Iba1 antibody. The accumulation of Iba1-positive cells was analyzed in three different brain regions, cerebral cortex, midbrain, and cerebellum, using immunofluorescence staining. (**A**,**B**) Representative images of three different brain regions from three independent WT mice or *Grn*^−/–^ mice. (**C**) Statistical analysis of A and B. Scale bar: 100 µm. *n* = 3. Data presented as mean ± SD. ** *p* < 0.01.

**Figure 3 ijms-23-00629-f003:**
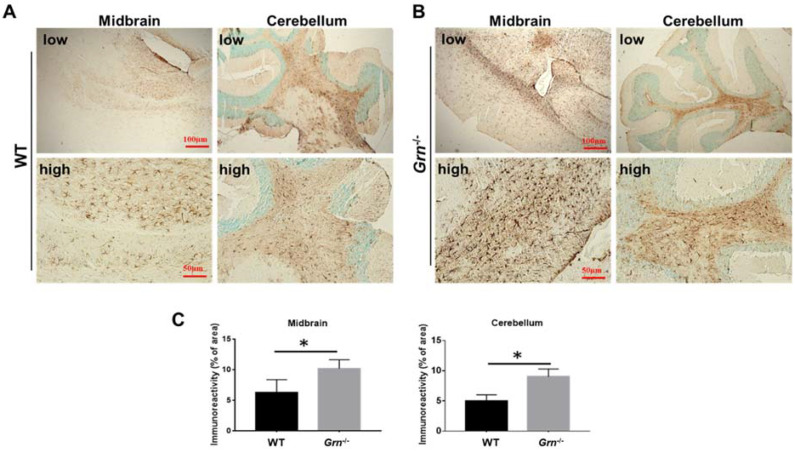
Ablation of PGRN promoted the activation of astrocytes in aged mice. Brain sections from 12-month-old WT and *Grn*^−/–^ mice were stained with anti-GFAP antibody. The accumulation of GFAP-positive cells was analyzed in two different brain regions, midbrain and cerebellum, using immunohistochemistry. (**A**,**B**) Representative images of three different brain regions from three independent WT mice or *Grn*^−/–^ mice. (**C**) Statistical analysis of A and B. *n* = 3. Data presented as mean ± SD. * *p* < 0.05.

**Figure 4 ijms-23-00629-f004:**
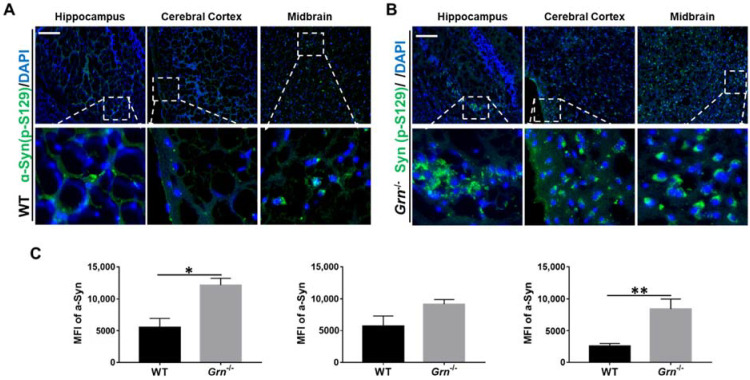
PGRN deficiency caused the accumulation of phosphorylated α-syn in aged mice. Frozen brain sections from 12-month-old WT and *Grn*^−/–^ mice were stained with anti-α-syn antibody. Brain regions of hippocampus, cerebral cortex, and midbrain were analyzed. (**A**,**B**) Representative images of three different brain regions from three independent WT mice or *Grn*^−/−^ mice. (**C**) The mean fluorescence intensity (MFI) of α-syn per section was quantified. Scale bar: 100 µm. *n* = 3. Data presented as mean ± SD. * *p* < 0.05, ** *p* < 0.01.

**Figure 5 ijms-23-00629-f005:**
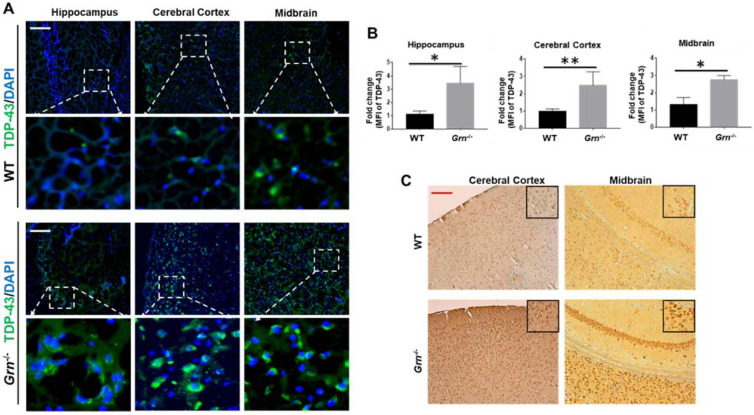
Accumulated TDP-43 in aged PGRN-deficient mice. Immunofluorescence and immunohistochemistry staining of TDP-43 in the brain sections from 12-month-old WT and *Grn*^−/–^ mice. (**A**) Representative images of TDP-43 accumulation in indicated brain regions by immunofluorescence staining. (**B**) Fold change of MFI of TDP-43 per section were quantified of A. (**C**) Immunohistochemistry staining of TDP-43 in the deparaffinized brain sections of 12-month-old WT and *Grn*^−/–^ mice. Scale bar: 100 µm. *n* = 3. Data presented as mean ± SD. * *p* < 0.05, ** *p* < 0.01.

**Figure 6 ijms-23-00629-f006:**
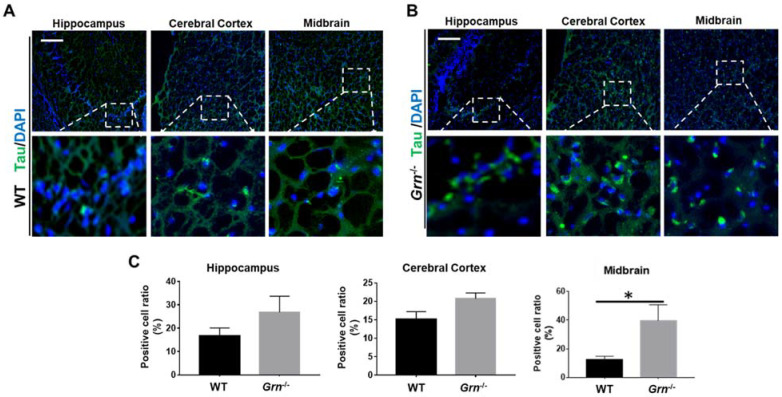
PGRN deficiency aggregated Tau in the CNS in aged mice. Tau antibody was used to stain the brain sections from 12-month-old WT and *Grn*^−/–^ mice. Distribution and accumulation of Tau positive cells were analyzed in three indicated brain regions using immunofluorescence staining. (**A**,**B**) Representative images of Tau accumulation in indicated brain regions of WT and *Grn*^−/–^ mice. (**C**) The ratio of Tau-positive cells per section were quantified of A and B. Scale bar: 100 µm. *n* = 3. Data presented as mean ± SD. * *p* < 0.05.

**Figure 7 ijms-23-00629-f007:**
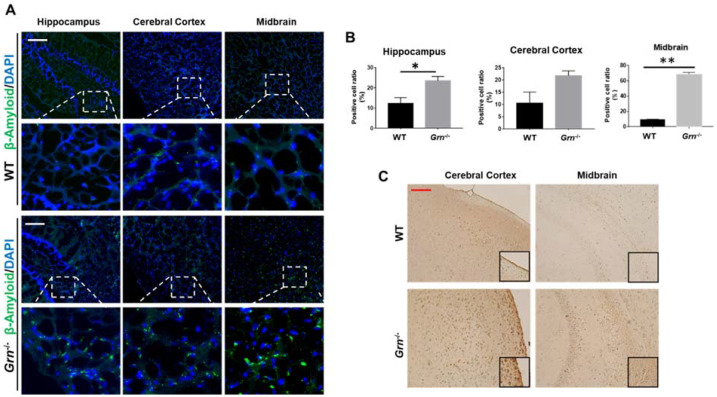
β-amyloid accumulated in aged PGRN-deficient mice. Immunofluorescence and immunohistochemistry staining of β-amyloid in the brain sections from 12-month-old WT and *Grn*^−/–^ mice. (**A**) Representative images of β-amyloid accumulation in indicated brain regions by immunofluorescence staining. (**B**) The ratio of β-amyloid-positive cells per section were quantified of A. (**C**) Immunohistochemistry staining of β-amyloid in the deparaffinized brain sections of 12-month-old WT and *Grn*^−/–^ mice. Scale bar: 100 µm. *n* = 3. Data presented as mean ± SD. * *p* < 0.05, ** *p* < 0.01.

**Figure 8 ijms-23-00629-f008:**
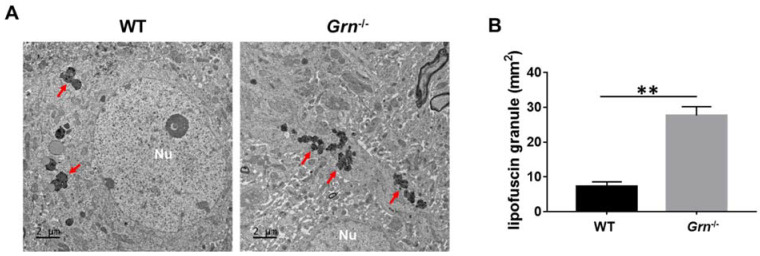
PGRN deficiency led to the accumulation of lipofuscin in a 17-month-old brain. Transmission electron microscopy was used to observe the lipofuscin granules in the brain of 17-month-old WT and *Grn*^−/–^ mice. (**A**) Lipofuscin granules in the brain tissue (red arrow) of WT and *Grn*^−/–^ mice. (**B**) Average number of lipofuscin granules detected per mm^2^. Nu: Nucleus. Scale bar: 2 µm. *n* = 3. Data presented as mean ± SD. ** *p* < 0.01.

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
