# Peer review of "Analysis of the Biomarkers for Neurodegenerative Diseases in Aged Progranulin Deficient Mice"

_ijms, 2022, doi:10.3390/ijms23020629_

Round 1

Reviewer 1 Report

I have examined the manuscript of Zhao et al., entitled 'Analysis of the biomarkers for neurodegenerative diseases in aged progranulin deficient mice' and I found it rigurous, with fine methodology and well written.

I recommend publication in its submitted form, since it brings new data in the field of neurodegeneration - regarding PGRN - which might be of significance.

Author Response

We thank Reviewer for the positive comments.

Reviewer 2 Report

This work is a well-thought-out, neatly implemented, and clearly presented study and can be published with minimal revisions. The logic is not entirely clear why the results of experiments on the effect of PGRN deficiency on autophagy are absent in the description of the results and are included in the supplementary section, despite the fact that they are a full-fledged part of the results and are described in the abstract and discussion. Typo on page 8 "dieseases".

Author Response

Reviewer 2:

This work is a well-thought-out, neatly implemented, and clearly presented study and can be published with minimal revisions. The logic is not entirely clear why the results of experiments on the effect of PGRN deficiency on autophagy are absent in the description of the results and are included in the supplementary section, despite the fact that they are a full-fledged part of the results and are described in the abstract and discussion. Typo on page 8 "dieseases".

Response: We thank Reviewer for the positive comments. We previously found that PGRN deficiency led to the defective of autophagy in Gaucher disease, a kind of lysosomal disease, so we checked if PGRN deficiency could affect autophagy in the brain (Please see the page 8, marked with Track Changes).  We have corrected the wrong writing (Please see the page 8, marked with Track Changes).

Reviewer 3 Report

This study is a comprehensive analysis of neurodegenerative disease markers in progranulin-deficient mice in aging, however without investigating mechanisms. It strengthens the role of progranulin as biomarker and its involvement in a wide array of diseases. Detection of early neuronal (NeuN) loss supports that progranulin is involved in the aging process and development of neurodegenerative diseases. The microglial (Iba1) analysis contributes to the, still not completely understood, role of progranulin in neurodegenerative diseases with a strong immune component such as Alzheimer’s disease (AD). It is interesting however, that b-amyloid is detected primarily in the midbrain, an area not strongly affected in AD. Other studies have detected microglial activation primarily in thalamus in progranulin-deficient mice (see for instance Zhang et al., Nature 2020).

Specific comments

In general, for Figures 4-7, the negative/WT immunofluorescent images appear to not show uniform cell staining (e.g. there are patterns of cell-void areas). Perhaps this a due to image processing.

Figure 2: The microglia (Iba1) analysis, uses positive cells for quantification. Microglial cells are normally expressing Iba1 and either MFI or % area covered may be more suitable.

Figure 4: Figure legend mentions that positive cells were quantified, however y axis says MFI.

Furthermore, the use of the expression positive cell ratio for y axis labeling seems misleading when percentage is used for the bars.

Author Response

We appreciate Reviewer for the positivity and recommendation below that further strengthen the paper. We also detected b-amyloid accumulation in thalamus, but not as much as in midbrain.

Specific comments

In general, for Figures 4-7, the negative/WT immunofluorescent images appear to not show uniform cell staining (e.g. there are patterns of cell-void areas). Perhaps this a due to image processing.

Response: Thanks for the comments. This is due to the selection area. The expression of the measured molecules in Figure 4-7 are very low in WT group, I select the area with expression to keep consistent.

Figure 2: The microglia (Iba1) analysis, uses positive cells for quantification. Microglial cells are normally expressing Iba1 and either MFI or % area covered may be more suitable.

Response: Thanks for the comments. As suggested, we re-analysis the results using MFI (Please see the page 4, the new figure 2).

Figure 4: Figure legend mentions that positive cells were quantified, however y axis says MFI.

Response: Thanks for the comments. As suggested, we corrected out statement (Please see the page 5, marked with Track Changes).

Furthermore, the use of the expression positive cell ratio for y axis labeling seems misleading when percentage is used for the bars.
Response: Thanks for the comments. We understand the reviewer’s concern; we did the analysis and show it according to some literatures (Chi-Chen Huang et al. Journal of Cell Science, 2014; Colleen M. Dewey et al. Molecular and Cellular Biology, 2011; Samuel M. Lee, BRAIN COMMUNICATIONS, 2019).